The effects of the gender-culture interaction on self-reports of depressive symptoms: cross-cultural study among Egyptians and Canadians

Huang Vivian 1
Beshai Shadi Shadi.Beshai@uregina.ca 2
Yu Mabel 2
1 Department of Psychology, Ryerson University , Toronto , Ontario , Canada
2 Department of Psychology, University of Regina , Regina , Saskatchewan , Canada
Kronfol Ziad
Electronic publication date: 2016 Dec 7
Publication date: 2016
Volume: 4
Electronic Location ID: e2783
Received 2016 Jun 28; Accepted 2016 Nov 10
Copyright: ©2016 Huang et al.
Copyright year: 2016
Copyright holder: Huang et al.
License: This is an open access article distributed under the terms of the Creative Commons Attribution License, which permits unrestricted use, distribution, reproduction and adaptation in any medium and for any purpose provided that it is properly attributed. For attribution, the original author(s), title, publication source (PeerJ) and either DOI or URL of the article must be cited.
License URL: https://creativecommons.org/licenses/by/4.0/

Keywords: BDI, Depression, Canada, Cognitive, Egypt, Culture, Somatic, Symptom profile

Funding: Canada Graduate Scholarship–Master’s Award Social Sciences and Humanities Research Council Doctoral Training Award Vivian Huang was supported by a Canada Graduate Scholarship–Master’s Award. Shadi Beshai was supported by a Social Sciences and Humanities Research Council Doctoral Training Award. The funders had no role in study design, data collection and analysis, decision to publish, or preparation of the manuscript.

==============================
Purpose

Research in depression has revealed differences in the way depressed individuals across cultures report their symptoms. This literature also points to possible differences in symptom reporting patterns between men and women. Using data from a larger dataset (Beshai et al. 2016), the current study examined whether non-depressed and depressed Egyptian and Canadian men and women differed in their self-report of the various domains of the Beck Depression Inventory –II (BDI-II).

Method

We recruited a total of 131 depressed and non-depressed participants from both Egypt (n = 29 depressed; n = 29 non-depressed) and Canada (n = 35 depressed; n = 38 non-depressed). Depression status was ascertained using a structured interview. All participants were asked to complete the BDI-II along with other self-report measures of depression. BDI-II items were divided into two subscales in accordance with Dozois, Dobson & Ahnberg (1998) factor analysis: cognitive-affective and somatic-vegetative subscales.

Results

We found a significant three-way interaction effect on the cognitive-affective (F(1,121) = 9.51, p = .003) and main effect of depression status on somatic-vegetative subscales (F(1,121) = 42.80, p < .001). Post hoc analyses revealed that depressed Egyptian men reported lower scores on the cognitive-affective subscale of the BDI-II compared to their depressed Canadian male counterparts.

Conclusions

These results suggest that males across cultures may differentially report cognitive symptoms of depression. These results also suggest that clinicians and clinical scientists need to further examine the interaction effect of culture and gender when investigating self-reported symptoms of depression.

Introduction

Cross-cultural epidemiological data indicate that depression is prevalent in almost all regions of the world (Kessler & Bromet, 2013). Specifically, epidemiological studies carried out in countries such as Lebanon (Karam et al., 2008), Morocco (Kadri et al., 2010), Saudi Arabia (Desouky, Abdellatif & Salah, 2015), and Egypt (Ghanem et al., 2009) reveal that depression is a prevalent condition in the Arab region. Depression carries an enormous global burden (Murray & Lopez, 1997). Therefore, studies that attempt to elucidate the nature and features of this condition worldwide are necessary. Cross-cultural research in the field of depression has revealed important culture-related differences in the experience and presentation of the disorder. For example, a number of early studies found that non-Western individuals suffering from depression may present with somatic features of the condition (e.g., appetite and sleep disturbance, agitation, etc.; Ryder, Yang & Heine, 2002). However, relatively more recent evidence suggests that Western patients specifically may “psychologize” their presentation of depression. In other words, the difference in depressive symptom presentations between Western and non-Western individuals may be largely driven by Westerners’ unique emphasis upon cognitive symptoms (e.g., feeling of sadness and anhedonia, negative thoughts about self, etc.; Kalibatseva & Leong, 2011; Parker, Cheah & Roy, 2001; Ryder et al., 2008). Due to this differential emphasis across cultures, this study attempted to examine differences between depressed Canadians and Egyptians in self-reported somatic and cognitive-affective symptoms of depression.

Somatization of depression has been observed in a number of cultures around the world. For example, South Asian immigrants to the United States are less likely than Americans of European decent to recognize and label a vignette based on emotional features as depression (Karasz, 2005). Similarly, Koss-Chioino (1999) found that Puerto Rican patients of depression generated fewer and less varied reports of cognitive symptoms in comparison to their reports of somatic symptoms.

In support of the somatization hypothesis among Middle-Eastern sufferers, Okasha (1999) indicated that, in describing their symptoms, depressed individuals in the Arab region reported shortness of breath, agitation, and other physiological changes. Further, a factor analytic study of the Centre for Epidemiologic Studies—Depression Scale (CES-D; Radloff, 1977) found that somatic and affective items of the scale, which are typically represented by different factors among Westerners, were fused in one factor among a sample of Egyptian students (Beshai, Dobson & Adel, 2013). El-Rufaie et al. (1999) found that depression and anxiety were both significantly related to somatic complaints, such as abdominal pain, among their Saudi Arabian samples. Further, Farhood & Dimassi (2012) found that depression, anxiety, and somatic complaints were significantly associated among a sample of Lebanese citizens experiencing war related post-traumatic stress symptoms. Finally, Simon and colleagues (1999) found that 51% of patients with depression had unexplained somatic symptoms, while 5% of this depressed sample denied the psychological symptoms of their depression.

By contrast, some recent evidence suggests that the concept of somatization should not be over-applied, and that this tendency is not entirely unique to non-Westerners. For example, somatization is often emphasized in Chinese populations due to the different cultural interpretations of the importance and/or the acceptability of somatic and psychological symptoms (Ryder, Yang & Heine, 2002). However, this characterization is not limited to the Chinese culture as evidenced by the findings of Zhou and colleagues (2011), who found that anxious Euro-Canadian participants were more likely than their Han Chinese counterparts to somatize their anxiety symptoms.

Further, emerging evidence suggests that the somatic emphasis in most regions of the world may be due to differences in symptom reporting patterns, as opposed to the nature of the disorder in these regions. For example, immigrant women from South Asia who lived in England recognized that depression is associated with both cognitive and somatic symptoms, but they emphasized the latter since they perceived them as more legitimate (Burr & Chapman, 2004). In Arab nations, cognitive symptoms, such as guilt, may be associated with “sin” in Islam, and thus their presence may imply moral culpability (Beshai, Clark & Dobson, 2013; El-Islam, 1969). Further, Goldberg & Bridges (1988) argued that somatization is a way for psychologically distressed individuals to assume the sick role without the stigma of mental illness. Past findings show that the stigma surrounding depression is more severe among Chinese Americans than European Americans (Hsu et al., 2008). As such, Chinese populations may present somatic symptoms more readily than Western populations in order to avoid psychiatric stigma.

Further, the distinction between symptom experience and reporting is not as straightforward as previously thought. Recent findings suggest that culture plays an important role in symptom presentation (e.g., Dere et al., 2013). For example, since the Chinese culture encourages a focus away from internal experiences (e.g., emotions) and towards concrete details of the external world, Chinese patients may perceive somatic symptoms as more salient than cognitive symptoms (Ryder & Chentsova-Dutton, 2012). In this regard, somatization is a culturally shaped style for expressing distress. Additionally, linguistic factors also contribute to the differential emphasis of somatic and cognitive symptoms across cultures. Although there are words to describe psychological states in the Chinese language, somatic metaphors are part of the communication style of the culture and thereby often used to express distress (Lee & Kleinman, 2007; Leff, 1988). Similarly, Hamdi, Amin & Abou-Saleh (1997) have suggested that the emphasis on somatic features among Arab sufferers of depression may be due to the lack of available linguistic idioms to express cognitive and affective symptoms in this region. As such, it is possible that translated Western questionnaires and other diagnostic instruments administered to participants in this region may provide a lexicon for such idioms.

Thoughts about self and future have often been associated with cognitive-affective symptoms of depression (Beshai, Clark & Dobson, 2013; Beshai et al., 2016). Therefore, negative self-referent thoughts have often been believed to be absent among depressed individuals of non-Western, and particularly Arabic origins (reviewed in Beshai, Dobson & Adel, 2012). Inconsistent with such beliefs, Beshai and colleagues (2012), Beshai and colleagues (2016) found that Egyptian individuals showing signs of depression reported similar levels of negative and positive thoughts about the self when compared with their Canadian counterparts.

Methodological artifacts may also explain some of findings in the literature regarding differences in self-reported depression symptoms across cultures. For example, some research suggests that individuals of Arabic descent express cognitive and affective symptoms of depression more readily when the mode of assessment takes the form of paper-and-pencil self-reports (Matthey, Barnett & Elliott, 1997). In addition, some evidence suggests that nonverbal reinforcement of cognitive-affective versus somatic responses by clinicians may be related to the preferential reporting of cognitive-affective symptoms in Western countries (Lam, Marra & Salzinger, 2005). As Chentsova-Dutton & Tsai (2008) argued, both somatic and cognitive features of depression are part of the disorder across cultures, and so the cultural emphasis of one set of symptoms over the other may reflect preferential reporting and comfort of disclosure rather than the experience itself. Although as mentioned above, it is difficult to disentangle symptom reporting from experience.

The majority of research in this area has investigated somatic and cognitive symptoms in broad categorical sets, rather than individual symptoms; however, recent findings highlight the need for a more nuanced examination (Dere et al., 2013). Parker, Cheah & Roy (2001) examined individual somatic and cognitive depression symptoms to determine the differences between Malaysian and Australian samples on the severity threshold for reporting certain depression symptoms. They found that although the actual prevalence of reporting “health problems” was comparable between the two groups (76% for Malaysian and 66% for Australian), the threshold at which the Malaysian sample reported this item was much lower than that of their Australian counterpart. Similarly, the severity threshold for reporting “chest pain” was also lower in the Malaysian sample, whereas the Australian sample reported such a symptom only when it was relatively severe. Studying symptoms at the individual level rather than amalgamating them into broad categorical sets may allow for a more nuanced understanding of the distinction between somatic and cognitive symptoms.

Recent findings by Dere et al. (2013) also attempted to address culture variations in symptom presentation through a more nuanced fashion. The researchers examined individual somatic and cognitive symptoms and found that certain symptoms deviated from their overall symptom reporting pattern. Despite reporting lower levels of cognitive symptoms compared to Euro-Canadian participants, the Chinese sample endorsed higher levels of affective symptoms (depressed mood, suppressed emotions) relative to their overall trend of reporting. Moreover, contrary to the popular assumption that Chinese are more reluctant to discuss depression, the Chinese and Euro-Canadian samples reported “depressed mood” at similar levels (56% and 48.6%, respectively). Further, and inconsistent with previous findings that Chinese outpatients endorsed significantly greater levels of somatic symptoms compared to Euro-Canadian outpatients (Ryder et al., 2008), Dere et al. (2013) found that Euro-Canadian participants had significantly greater levels of atypical somatic symptoms—appetite gain, weight gain, and hypersomnia—than their Chinese counterpart. Although the literature provides substantial evidence for non-Western emphasis on somatization and Western emphasis on psychologization, these findings suggest that this generalization may not hold true across all symptoms.

In addition to cross-cultural differences, there are possible gender differences in the experience of depressive symptoms. Consistent findings suggest that women are twice as likely as men to experience depressive in their lifetime (Angst et al., 2002; for review, see Kuehner, 2003). Further, these gender differences in the prevalence of depression appear to be consistent across cultures (Maier et al., 1999). Previous studies have elucidated various underlying factors that may contribute to gender differences in depression rates, which include cognitive (e.g., Nolen-Hoeksema & Jackson, 2001), biological (e.g., Staley et al., 2006; Moieni et al., 2015), neurological (e.g., Yao et al., 2014), and social (e.g., Dalgard et al., 2006) factors. Consistent with literature from the West, a review by Eloul, Ambusaidi & Al-Adawi (2009) found heightened rates of depression among women in Middle Eastern and North African countries compared to men.

In addition to differences in prevalence, one study suggested that females tend to report more somatic and less cognitive symptoms of depression and when compared with their male counterparts (e.g., Schuch et al., 2014). Specifically, women appear to more frequently report changes in appetite, sleep disturbance, psychomotor retardation, negative perception of physical health, and greater functional impairment than do men (Kornstein et al., 2000; Marcus et al., 2005; Silverstein, 2002). Although some cross-cultural studies have examined different symptom reporting across genders, the focus has mainly been on females and somatization of symptoms. Very few cross-cultural studies to date have examined gender differences in reporting both cognitive and somatic depressive symptoms. Thus, studies that examine cross-cultural and cross-gender differences in symptom reporting in depression may be worthwhile.

Understanding differential experiences and reporting of symptoms across cultures is important, as this understanding may aid in tailoring effective treatments for individuals of varying cultural groups. For example, in cognitive therapy for depression (Beck et al., 1979), which is one of the most used and evaluated psychotherapies for the disorder, a significant portion of therapeutic time and effort is spent tackling “cognitive” features of the disorder (negative automatic thoughts, attitudes, etc.; Beck et al., 1979). Accordingly, if these symptoms are not routinely reported or if they do not significantly correlate with meaningful depressive experience among people of Arabic descent, then cognitive therapy’s dedication to dismantling negative cognitions may be unwarranted among these patients. Further, examining the effects of the culture-gender interaction on self-reported symptoms of depression may aid in the appropriate and more nuanced assessment and diagnosis of this disorder in various cultures.

The current study attempted to close the gaps in the current literature by examining the frequency and relationships of somatic and cognitive-affective symptoms of depression among a group of depressed Egyptian patients. The data for this study were collected as part of a larger trial (Beshai et al., 2016) that examined the cognitive model of depression in Egypt. In the current study, we computed separate scores for the somatic-vegetative and cognitive-affective domains of the Beck depression Inventory-II (BDI-II; (Beck, Steer & Brown, 1996), and compared these cross-culturally (Egyptians and Canadians), across genders, and across depression statuses (depressed and non-depressed). In accordance with earlier studies on somatization among Egyptians (e.g., Okasha, 1999), we hypothesized a culture by depression status interaction on the somatic-vegetative subscale of the BDI-II, wherein depressed Egyptians would exhibit higher scores on this subscale than depressed Canadians. Given that the literature suggests Westerners tend to “psychologize” their depression (Ryder, Yang & Heine, 2002), we hypothesized a second culture by depression status interaction, such that, in comparison to depressed Egyptians, depressed Canadians would show higher scores on the cognitive-affective subscale of the BDI-II. We did not anticipate a three-way interaction.

Method

Data from a total of 58 Egyptians (29 depressed and 29 non-depressed) and 73 Canadians (35 depressed and 38 non-depressed) were used for this study. The data were collected in the context of a larger trial by Beshai and colleagues (2016). Depressed participants were either recruited from a psychiatric clinic (Cairo, Egypt), or from the community via poster notices and social media (e.g., Facebook) advertisement. Non-depressed Egyptian participants were recruited from the community at large, whereas non-depressed Canadians were recruited via recruitment posters or social media outlets that identified the target sample (e.g., individuals with depression) in the community of Calgary, Alberta, Canada. A power analysis was conducted to estimate the appropriate sample size for the current study. Using G*Power (Faul et al., 2007), a minimum sample size of 111 was needed to detect a medium–large effect (Cohen’s f = 0.35; with a priori α = .05, β = .80) based on the current 2 × 2 × 2 design.

To determine eligibility of potential participants, study researchers administered a bespoke structured interview (Structured Interview Screen, described below) designed to ascertain the presence of depressive symptoms and absence of other exclusionary diagnostic criteria (e.g., mania, psychosis, substance dependence). All depressed participants in both countries exhibited symptoms in accordance with DSM-IV-TR (American Psychiatric Association, 2000) Major Depressive Episode criteria. All non-depressed participants showed no diagnostic criteria for bipolar disorder and other aforementioned exclusionary diagnoses, and had no previous history of major depression. Eligible participants were then invited to the clinic/lab to participate in the study. Upon arrival, all participants provided written informed consent and completed a questionnaire package, which included demographic information and the BDI-II (Beck, Steer & Brown, 1996) Depression status was ascertained through responses on the initial structured interview, a score of 14 or higher on the BDI-II, and a score of 9 or higher on the Psychiatric Diagnostic Screening Questionnaire Depression Subscale (Zimmerman & Mattia, 2001). The study was approved by the Conjoint Faculties Research Ethics Board (CFREB) at the University of Calgary (Approval reference: REB-6899).

To be included, participants reported that Egypt (for the Egyptian sample) or Canada (for the Canadian sample) is the country of birth of self and parents, and that they were at least third generation Egyptian (Egyptian sample) or third generation European (Canadian sample). Finally, participants with symptoms of psychosis, mania, and substance dependence were excluded from the study.

The Structured Interview Screen was initially administered to the Canadian and Egyptian samples to determine the eligibility of participants. The screen consists of 14 questions, five of which were pertaining to current depressive symptoms. The remaining items assessed symptoms related to diagnostic (i.e., psychosis, mania, and substance dependence) and demographic (e.g., country of birth) exclusionary criteria. The items were constructed based on the Structure Clinical interview for the DSM-IV (SCID; First et al., 1997) and the Composite International Diagnostic Interview 3.0 (Kessler & Ustun, 2004). The depression items had good reliability (α = .88) within the Canadian sample. Moreover, Canadian participants who reported “Yes” to the interview question “In the last month, did you feel sad or depressed for most of the day, nearly everyday” scored significantly higher on the BDI-II than those who answered “No” to this question (BDI-II: F(2, 82) = 115.22, p <.001). Accordingly, this screening interview appeared to have good reliability and appeared consistent with other depression measures in differentiating between depressed and non-depressed participants.

The Beck Depression Inventory –II (BDI-II; Beck, Steer & Brown, 1996) is a 21-item self-report measure that evaluates the severity of depressive symptoms. The scale contains various domains of the disorder, which include affective, cognitive, somatic, and motivational domains. Items are answered on a 4-point Likert scale (ranging from 0 to 3), with a higher score on the scale indicating greater severity of depressive symptoms. In order to operationalize the somatic and cognitive symptom domains of depression, previous studies have conducted exploratory and confirmatory factor analyses on the BDI-II to identify its latent factors (Dozois, Dobson & Ahnberg, 1998; Steer et al., 1999; Whisman, Perez & Ramel, 2000). Specifically, a confirmatory factor analysis conducted by Dozois and colleagues (1998) suggested that items 1, 2, 3, 5, 6, 7, 8, 9, 13, and 14 were representative of the cognitive-affective domain and items 4, 10, 11, 12, 15–21 were representative of the somatic-vegetative domain of the BDI-II within a college sample. Furthermore, Dozois and colleagues (1998) found that the two latent factors of the BDI-II were highly correlated. The current study calculated somatic-vegetative and cognitive-affective symptom subscales in accordance with domains identified by Dozois, Dobson & Ahnberg (1998). A series of ordinal logistic regressions were conducted to establish cross-cultural equivalence of the BDI-II and the aforementioned subscales (See detailed analysis plan below).

All the materials used in Egypt were translated according to the World Health Organization’s guidelines. Accordingly, two professional translators were involved in the translation process of study material: one translator who forward translated the English materials into Arabic, and then the other who back translated the Arabic translation back to English.

All analyses were performed on SPSS 23.0 (IBM, Chicago, USA). An alpha level of 0.05 was used to determine statistical significance. In order to establish scalar equivalence, a series of ordinal logistic regressions were conducted to assess the uniform and non-uniform differential item functioning across the two ethnicities of participants. The analyses were conducted separately for the BDI-II total score, the BDI-II cognitive-affective and somatic-vegetative subscale scores. Following Zumbo (1999) and Slocum, Gelin & Zumbo (2003), each item was entered as the independent variable. Dependent variables were entered in three separate steps: (1) total score, (2) cultural group variable, (3) total score × cultural group. Two criteria were set to identify differential functioning of items: (a) significant two-degree-of-freedom chi-square test (p<.01), and/or (b) a change in Nagelkerke R2 of .13 or larger between Steps 1 and 3 (uniform differential functioning), and between Steps 2 and 3 (non-uniform differential functioning). If an item displayed differential functioning, it was removed from subsequent analyses.

All differences on key demographic variables between the Egyptian and Canadian samples were assessed using one-way analysis of variance (ANOVA) and chi-square analysis. Using the scalar invariant items, a three-way (Nationality by Gender by Depression status) ANOVA was conducted in order to examine cross-national and gender differences in total BDI-II scores Pearson correlation coefficients were calculated to examine the relationships between somatic-vegetative and cognitive-affective symptoms, as well as the total score of the BDI-II within each subsample (i.e., Egyptian vs. Canadian). Two, three-way analyses of covariance (ANCOVAs) were conducted in order to test cultural, gender, and depression status differences on the BDI-II somatic-vegetative and cognitive-affective domains, while controlling for the other subscale score. Finally, independent samples t-tests were conducted post hoc to delineate any significant interactions found in the ANCOVA analyses.

Results

Participant characteristics

Participant had a mean age of 33.94 years (SD = 13.37; range = 18–65). See Table 1 for detailed sample characteristics. A one-way ANOVA revealed that Canadians were significantly older than their Egyptian counterparts, F(1, 129) = 8.83, p = .004. The chi-square analyses revealed that there were significant differences in economic status (χ2[5] = 13.28, p = .02) and education level (χ2[8] = 21.11, p = .007) between the Egyptian and Canadian samples. Age was also found to significantly correlate with other depression-related constructs (e.g., hopelessness; Beshai et al., 2016). Accordingly, age was used as a covariate in subsequent ANCOVAs.

Table 1 Sample characteristics of the Egyptian and Canadian samples.

	Egyptian (N = 58)	Canadian (N = 73)	
	Male (n= 24)	Female (n= 34)	Male (n= 20)	Female (n= 53)	
	Depressed (n = 12)	Non-depressed (n = 12)	Depressed (n = 17)	Non-Depressed (n = 17)	Depressed (n = 12)	Non-depressed (n = 8)	Depressed (n = 23)	Non-depressed (n = 30)	
Age:	
Mean (SD)	26 (7.15)	34.08 (11.64)	31.82 (8.64)	28.65 (11.26)	38.58 (15.00)	36.38 (16.43)	42.65 (12.68)	32.07 (15.09)	
Economic status (%):	
Unemployed/No yearly income	7 (58.3)	3 (25.0)	8 (47.1)	13 (76.5)	4 (33.3)	2 (25.0)	4 (17.4)	8 (26.7)	
10,000–50,000	4 (33.3)	9 (75.0)	2 (11.8)	2 (11.8)	6 (50.0)	2 (25.0)	12 (52.2)	15 (50.0)	
50,000 and over	1 (8.3)	0	5 (29.4)	1 (5.9)	2 (16.7)	3 (37.5)	4 (17.4)	6 (20.0)	
None of the Above	0	0	2 (11.8)	1 (5.9)	0	1 (12.5)	3 (13.0)	1 (3.3)	
Education (%):	
Secondary school/Below	4 (33.3)	4 (33.3)	2 (11.8)	7 (41.2)	6 (50.0)	4 (50.0)	4 (17.4)	13 (43.3)	
Trades certificate/Diploma	1 (8.3)	6 (50.0)	3 (17.6)	3 (17.6)	1 (8.3)	1 (12.5)	11 (47.8)	7 (23.3)	
Bachelor’s Degree/Above	7 (58.3)	2 (16.7)	12 (70.6)	7 (41.2)	5 (41.7)	3 (37.5)	8 (34.8)	10 (33.3)	
Marital status (%):	
Single, never married	10 (83.3)	5 (41.7)	11 (64.7)	12 (70.6)	7 (58.3)	4 (50.0)	12 (52.2)	18 (60.0)	
Married	2 (16.7)	7 (58.3)	4 (23.5)	5 (29.4)	4 (33.3)	3 (37.5)	3 (13.0)	10 (33.3)	
Separated/Divorced	0	0	2 (11.8)	0	1 (8.3)	1 (12.5)	8 (34.8)	2 (6.7)	
Religion (%):	
Christianity	8 (66.7)	8 (66.7)	15 (88.2)	9 (52.9)	6 (50.0)	4 (50.0)	14 (60.9)	18 (60.0)	
Islam	3 (25.0)	4 (33.3)	2 (11.8)	8 (47.1)	0	0	0	0	
Agnostic/Atheist	0	0	0	0	3 (25.0)	3 (37.5)	5 (21.7)	7 (23.3)	
Other	1 (8.3)	0	0	0	3 (25.0)	1 (12.5)	4 (17.4)	5 (16.7)	
BDI-II Total scorea (SD)	20.21 (6.52)	3.04 (3.02)	27.53 (8.38)	4.53 (3.52)	28.75 (10.17)	1.25 (1.28)	30.00 (8.53)	2.50 (2.43)	
BDI-II Somatic-vegetative symptomsa	11.96 (4.39)	1.21 (1.34)	13.59 (5.07)	2.35 (3.10)	14.42 (5.63)	0.75 (0.89)	16.57 (4.03)	1.57 (1.59)	
BDI-II Cognitive-affective symptomsa	8.25 (4.02)	1.83 (2.21)	13.94 (4.59)	2.18 (3.52)	14.33 (5.26)	0.50 (0.53)	13.43 (5.14)	0.93 (1.53)	
Notes.

a New calculated BDI-II total scores with cultural variant items removed.

Cross-cultural equivalence of the BDI-II

The three-step ordinal logistic regression analyses revealed that items 6 (Punishment feelings) and 11 (Agitation) displayed significant differential functioning. Specifically, the analyses revealed that item 6 showed a significant 2-df chi-square on total BDI-II scores (χ2 = 21.56, p < .01; ΔNagelkerke R2 = 0.13) and on the cognitive-affective subscale (χ2 = 19.91, p < .01; ΔNagelkerke R2 = 0.12). Similarly, item 11 displayed a significant chi-square on total BDI-II scores (χ2 = 12.74, p = .002; ΔNagelkerke R2 = 0.06) and on the somatic-vegetative subscale (χ2 = 9.66, p = .007; ΔNagelkerke R2 = 0.03). These items were removed from subsequent analyses. The remaining 19 items of the BDI-II evidenced cross-cultural scalar invariance, and thus were retained in calculating total and subscale scores. The final BDI-II total and subscale scores displayed excellent internal consistencies among Egyptian (total: α = .93; cognitive-affective: α = .88; somatic-vegetative: α = .90) and Canadian samples (total: α = .97; cognitive-affective: α = .94; somatic-vegetative: α = .95).

Cross-national, gender, and depression status differences on total BDI-II scores

A three-way ANOVA was conducted to examine cross-national, gender, and depression status differences on total BDI-II scores. The analysis revealed a significant nationality by depression status interaction (F(1, 123) = 9.87, p = .002, partial η2 = .07), whereby depressed Canadians reported higher total scores on the BDI-II compared to depression Egyptians. Among non-depressed participants, there was no significant difference between nationalities on total scores of the BDI-II. There was also no significant nationality by gender interaction evidenced in the current analysis (F(1, 123) = 1.79, p = .18), nor a gender by depression status interaction (F(1, 123) = 1.53, p = .22). However, a significant main effect of gender on total BDI-II scores was revealed, F(1, 123) = 5.76, p = .02 partial η2 = .05). Specifically, females reported significantly higher total BDI-II scores than did males.

Correlational analyses of total and subscales BDI-II scores between Egyptians and Canadians

A Pearson correlation analysis evidenced that BDI-II total scores were significantly and positively correlated with the BDI-II somatic-vegetative symptoms among Egyptians (r(57) =.94, p < .001) and Canadians (r(72) = .98, p < .01). Similarly, scores on the BDI-II cognitive-affective subscale were significantly and positively correlated with total BDI-II scores among Egyptians (r(57) = .93, p < .001) and Canadians (r(72) = .98, p < .001) Scores on the BDI-II somatic-vegetative and cognitive-affective subscales were significantly and positively correlated with each other among Egyptians (r(57) = .76, p< .001) and Canadians (r(72) = .92, p < .01). However, a Fisher’s r-to-z transformation analysis revealed a stronger relationship between the somatic-vegetative and cognitive-affective subscales among Canadians than among Egyptians (Fisher’s z = 3.29, p = .001).

Nationality, gender, depression status and cognitive-affective subscale scores

The three-way, 2 (Egyptian vs. Canadian) by 2 (Female vs. Male) by 2 (Depressed vs. Non-depressed) ANCOVA was conducted in order to examine cross-cultural, gender, and depression status differences on the BDI-II cognitive-affective and somatic-vegetative subscales, using age and either subscale score as covariate when analyzing the other as the dependent variable. The first analysis revealed a significant three-way interaction on the cognitive-affective subscale of the BDI-II F(1,121) = 9.51, p = .003, partial η2 = .07 (See Fig. 1) This analysis also evidenced a significant depression status by nationality interaction (F(1, 121) =4.87, p = .03, partial η2 = .04) and a significant nationality by gender interaction (F(1, 121) = 8.047, p = .005, partial η2 = .06) on the cognitive-affective subscale.

Figure 1 Three-way interactions of BDI-II cognitive-affective subscale scores.

(A) Depressed individuals. Covarying for age and somatic-vegetative subscale scores; and (B) non-depressed individuals. Covarying for age and somatic-vegetative subscale scores.

As planned, follow-up independent sample t-tests were used to tease apart the significant interaction found in the cognitive-affective domain of the BDI-II. Using Bonferroni corrections, the alpha level was corrected to .01 to account for family-wise error rate (FWER). The tests revealed that depressed Canadian males (M = 14.33, SD = 5.26) reported significantly higher scores on the cognitive-affective subscale of the BDI-II compared to depressed Egyptian males (M = 8.25, SD = 4.24), t(22) = 3.13, p = .005. There was a marginally significant cross-national difference in cognitive-affective symptoms of the BDI-II among non-depressed females. Specifically, non-depressed Egyptian females (M = 2.58, SD = 1.94) exhibited greater cognitive-affective symptoms on the BDI-II than their Canadian counterpart (M = 0.93, SD = 1.53), t(27.34) = −2.27, p = .03. There were no significant differences between depressed Egyptian and Canadian females (t(38) = −0.32, p = .75), and non-depressed Egyptian and Canadian males (t(12.86) = −1.99, p = .07).

Nationality, gender, depression status, and somatic-vegetative subscale scores

Controlling for age and the cognitive-affective subscale scores, the ANCOVA revealed a marginally significant three-way interaction of the somatic-vegetative subscale of the BDI-II, F(1,121) = 3.28, p = .07. Furthermore, there was a significant main effect of depression status (F(1, 121) = 42.80, p < .001, partial η2 = .03). By examining the means, depressed individuals (M = 14.51, SD = 4.91) reported significantly higher scores on the somatic-vegetative subscale compared to their non-depressed counterparts (M = 1.60, SD = 2.02).

Discussion

This study examined cross-cultural and gender differences on BDI-II subscale scores (somatic-vegetative and cognitive-affective symptoms) among depressed and non-depressed individuals. The results obtained in this study stress the need to examine depressive symptoms in non-Western cultures as a multifaceted construct. Further, the results indicated that men and women with different depression statuses in different cultures may report a different pattern of symptoms. Contrary to the hypothesis, there were no significant two-way interaction between nationality and depression status on the somatic-vegetative subscale of the BDI-II. However, there was a significant three-way interaction, wherein depressed Egyptian males reported less cognitive-affective symptoms than depressed Canadian males, after controlling for age and somatic-vegetative subscale scores. The current findings provided some new insight into cultural differences in depressive symptom presentation between Egyptians and Canadians.

This study also examined the scalar invariance of the BDI-II across cultures. These equivalence analyses revealed that, with the exception of the Punishment feelings (6) and Agitation (11) items, the majority of BDI-II items evidenced scalar equivalence across cultures. This suggests that the employed Arabic translation of the BDI-II was a valid measure of depression amongst this Egyptian sample. Previous studies have also found measurement invariance of the BDI-II in other cultures, such as Chinese (Byrne et al., 2007; Whisman et al., 2012), Turkish (Canel-Cinarbas, Cui & Lauridsen, 2011), and South African (Makhubela, 2016) among clinical and non-clinical samples. Given that this is one of the first studies to examine scale equivalence of the Arabic version of the BDI-II, future studies are needed to replicate the current findings using a larger sample, especially among depressed Egyptians. Future studies also need to explore other indicators of scale invariance (e.g., configural and metric invariance; Meredith, 1993).

Previous studies have suggested that non-Western cultures tend to emphasize somatic as opposed to cognitive, symptoms in relation to their depression (Chentsova-Dutton & Tsai, 2008; Koss-Chioino, 1999). This emphasis upon somatic complaints as key features of depression has also been found among South Asian immigrants (Karasz, 2005), and African-American patients (Sellers, Ward & Pate, 2006). Even in Egypt, some research has found that depressed individuals in that region of the world tend to emphasize somatic complaints (Okasha, 1999; Sami & El-Gawad, 1995). However, the present study found no significant differences on self-reports of somatic symptoms between depressed Egyptians and Canadians, after controlling for age and the cognitive-affective subscale scores. As suggested by Dere et al. (2013), the non-Western emphasis upon somatization should not be over generalized. However, it is possible that the current sample may not be fully representative of the population of depression sufferers in Egypt. Particularly, all depressed Egyptian participants were recruited from a psychiatric clinic in an affluent neighbourhood in Cairo, Egypt. Thus, these treatment-seeking individuals may experience less self-stigma and may have greater acceptance of their condition, which in turn may explain their relatively diminished emphasis on somatic symptoms. Furthermore, depressed Egyptians may also have experienced less depressive symptoms overall, as this was reflected in total BDI-II scores. As suggested by Ryder et al. (2008), more research is needed to examine the effect of help-seeking strategies on symptom presentation. Future studies should recruit Egyptians with depression via different avenues and/or methods to ensure the accurate representation of depression in this region of the world. Nonetheless, the present findings may shed light on depressive symptom endorsement between depressed Egyptians and European Canadians.

We hypothesized that depressed Canadians would exhibit significantly higher scores on the cognitive-affective subscale of the BDI-II. Partially consistent with our hypothesis, the present study found a significant three-way interaction on the cognitive-affective subscale of the BDI-II. Specifically, it was found that depressed Egyptian males reported significantly less cognitive-affective complaints in comparison to depressed Canadian males. Consistent with previous findings, Western individuals may emphasize psychological features of depression (i.e., greater emphasis on the cognitive-affective symptoms) compared to individuals of non-Western descent (e.g., Kalibatseva et al., 2014; Parker, Cheah & Roy, 2001; Ryder et al., 2008). However, this pattern of results also suggests differential symptom expression of depression among males in particular. Although there are limited studies that have attempted to examine the experience and expression of depression in men, the few studies that have been conducted suggest that male presentation might not map well onto existing diagnostic tools or self-report measures of depression (e.g., Cochran & Rabinowitz, 2000; for review see Addis, 2008). Moreover, stigma and gendered sociocultural norms in Egypt may have defined how men should think and behave. In turn, these norms may shape how men respond to the experience of depression and how they may express its symptoms (Addis & Cohane, 2005). Considering Egypt is a relatively conservative country (Okasha, 1999), it is possible that Egyptian males are adherent to these sociocultural and gender norms compared to their Canadian counterparts.

In addition to gender and cultural norms, linguistic differences between Arabic and English may result in different interpretations of the questionnaire items, especially cognitive items (Hamdi, Amin & Abou-Saleh, 1997). However, due to the paucity of research on depression and males specifically, it is difficult to identify specific factors that may contribute to the current findings.

Overall, females reported greater total depressive symptoms compared to their male counterparts. Consistent with previous findings, females are at greater risk of experiencing depressive symptoms due to various underlying factors (e.g., Dalgard et al., 2006; Moieni et al., 2015). Further, we did not find any gender by depression status interaction on total BDI-II scores. It appeared that, regardless of depression status, females tend to report greater overall depressive symptoms in the current sample. Contrary to our hypothesis, we found a significant nationality by gender interaction on the cognitive-affective subscale, wherein non-depressed Egyptian females reported higher scores on this subscale compared to their Canadian counterparts. Although differences were found, the mean cognitive-affective scores of the non-depressed Egyptian females were below the cutoff for minor depression (Alansari, 2006). The relatively elevated scores among non-depressed Egyptian females may be contributed to daily or social stressors, which have been associated with fluctuations in depressive symptoms (e.g., Hankin, 2010).

The pattern of correlations observed across cultures suggested that the nomological network of the depression construct may differ depending on the studied population (Cronbach & Meehl, 1955). For instance, although scores on BDI-II somatic-vegetative and cognitive-affective symptoms were correlated among Egyptian participants, the relationship was not as strong as that evidenced among Canadians. As suggested by Brouwer, Meijer & Zevalkink (2013), the existence of a strong association between somatic and cognitive symptoms among Western individuals suggested that both subconstructs were not orthogonal and may be driven by a higher order construct of depression. However, the differences in the strength of these correlations may be attributable to the samples sizes between the cultural groups, whereby the Canadian sample was slightly larger than the Egyptian sample. Nonetheless, it is possible that the somatic and cognitive symptoms represent orthogonal constructs within the current sample of Egyptians. Given the relatively simple design of this study and the modest sample size employed, this hypothesis remains to be explored in the future.

This study extends previous research in many ways. First of all, this study employed a secondary sample recruited from Canada to act as a cultural control group. Moreover, the use of a non-depressed sample also provided additional baseline information. Second, special care and attention went into the translation of measures used in this study (see Beshai et al., 2016, for details regarding the translation process). As a testament to the success of the translation process, the measures used in the current study possessed excellent and similar reliability estimates across samples, and evidenced partial construct validity. Moreover, scalar equivalence analyses were conducted to ensure that the BDI-II scores were valid across cultures. By removing items that displayed cultural variance, the differences found in this study were likely to reflect meaningful and real cross-cultural differences in the construct of depression. The measures used herein are globally used self-report measures of depression, and so the results obtained in this study can be directly compared with a large body of literature arising from different parts of the world.

Despite these strengths, the study suffered from a number of limitations. First of all, depressed Egyptians and Canadians were found to differ on a number of important characteristics (age, socioeconomic status, depression scores), and so the observed differences in symptom profiles may be due to these unexamined demographic or diagnostic differences. However, using the same dataset, Beshai et al. (2016) found that, besides chronological age, depressive symptoms and other depression-related constructs were not systematically related to any of the examined demographic factors. Second, along with cut-offs on self-report measures, the current study relied on a bespoke structured interview designed in accordance with DSM-IV criteria for major depressive episode to ascertain depression diagnoses. As such, this interview was not validated for this purpose. With this said, the interview items were reliable and significantly predicted scores on self-report measures. Further, most “gold standard” interviews, such as the Structured Clinical Interview for the DSM-IV (First et al., 1997), have never been translated to Arabic, and so their use in this study was not feasible. Third, the current study relied on scores of only one measure to examine self-reported differences across cultures in somatic and cognitive symptoms. Fourth, the sample size in the current study was relatively modest. This modest sample size makes generalizations to the population of Egyptian depression sufferers difficult. Fifth, the sample recruited in Egypt for the purposes of this study may not be fully representative of the general population of Egyptians. This is especially true given that all participants self-selected into the study thus this may have skewed the presented results. Finally, depression in this study was defined in accordance with DSM-IV criteria, which is a Western conceptualization of the disorder. Accordingly, it is possible that Western definitions of the depression do not map well onto this construct in Egypt, which may partially explain the differential findings. With these limitations in mind, the present study still answered a number of questions regarding self-reported symptoms of depression cross-culturally.

Given these limitations, future research should employ multiple methodologies (e.g., face-to-face interviews; self and clinician reports of symptomology) to examine the presentation of somatic and cognitive features of depression in Egypt. Further, and considering cross-cultural and gender differences obtained here, future research should employ larger samples of men and women in order to replicate the findings of the present study. Depression and anxiety are often comorbid and highly overlapping diagnostic construct (Clark & Watson, 1991). Therefore, future studies should examine whether there is agreement between reported symptom profiles of anxiety and depression cross-culturally Hoge et al. (2006).

Conclusions

Although both somatic and cognitive symptoms of depression are typically self-reported among individuals of Egyptian descent, depressed Egyptian men may be reluctant to report cognitive-affective symptoms in comparison to depressed men of Western descent. Further, it is possible that cognitive and somatic symptoms represent orthogonal subconstructs of depression in Egypt, and so it may be appropriate to measure these symptoms separately for individuals of this culture.

Supplemental Information

Data S1 Cross-cultural Depressive Symptoms Dataset

Click here for additional data file.

Additional Information and Declarations

Competing Interests

Author Contributions

Human Ethics

Data Availability

The authors report no conflict of interest.

Vivian Huang analyzed the data, contributed reagents/materials/analysis tools, wrote the paper, prepared figures and/or tables, reviewed drafts of the paper.

Shadi Beshai conceived and designed the experiments, performed the experiments, analyzed the data, contributed reagents/materials/analysis tools, wrote the paper, prepared figures and/or tables, reviewed drafts of the paper.

Mabel Yu wrote the paper, reviewed drafts of the paper.

The following information was supplied relating to ethical approvals (i.e., approving body and any reference numbers):

The University of Calgary Conjoint Faculties Research Ethics Board (CFREB; File number: REB-6899).

The following information was supplied regarding data availability:

The raw data has been supplied as a Data S1.

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
