# Peer review of "The effects of the gender-culture interaction on self-reports of depressive symptoms: cross-cultural study among Egyptians and Canadians"

_PeerJ, doi:10.7717/peerj.2783_

## Round 0.1 · original submission · Major Revisions

Your article received mixed reviews, although on balance they tend to be quite supportive. I therefore encourage you to resubmit your manuscript after major revisions. If you decide to resubmit, please address all the issues raised by all the reviewers. Please pay special attention to reviewer 3. In particular, the issue of cross-cultural equivalence, the power analysis and the problem of variance shared between the two subscales need to be clearly and convincingly addressed in your revision.

Reviewer 1 ·

Basic reporting

The report is well written and comprehensive. The literature needs to be updated with more recent studies from the Arab region related to somatic expression of depression, in countries such as Lebanon, Palestine, Iraq, United Arab Emirates etc...

Experimental design

The design is quite clear, with defined hypotheses based on literature. The sample is small in all groups which limits the significance and generalizability of results. It was explained that this is part of a larger study, trial that examined the model of depression.
Another aspect which researchers did not consider is to WHOM the depression complaint is being told, whether to a westerner or an arab. This will make a difference since social desirability to fit into any criteria of depression may alter the patients presentation, based on what he/ she thinks is expected to say. This variable was not taken into account yet may have played a significant role.
In addition the sample is a self selected sample which in itself creates a bias.

Validity of the findings

Some findings were not statistically significant due to the small sample number. The few results that were need to be better explained and discussed from a literature perspective,

·

Basic reporting

The structure of the paper is adequate. The language is clear. The introduction is a bit lengthy and could be shortened.

Experimental design

The experimental design is adequate. It is extrapolated from a well designed and conducted database as the authors desribe.

Validity of the findings

The data analysis is statistically sound. The conclusions drawn are based on the statistical findings and no speculation is made.

·

Basic reporting

1. There are several typos throughout. For example: ‘Dozios’ in the abstract should be ‘Dozois’; ‘post hoc’ should not be hyphenated; on line 49, “a number of early studies have found” should be “a number of early studies found”; on line 264 there should be no leading zero for the alpha level. There may be others. The manuscript should be carefully proofread before any resubmission.

2. Also, I recommend using ‘European Americans’ rather than ‘Caucasian Americans’, as the latter requires use of an outdated racial term.

Experimental design

1. One of the hypotheses involves a null finding, which should not really be done. The authors could soften the language here, to say that they did not anticipate any statistically significant differences. The authors should comment on whether they had sufficient power to find a meaningful difference, had one been present.

Validity of the findings

1. My main concern about validity involves the lack of attention to cross-cultural equivalence. How do we know that, for example, a given BDI-II subscale is understood the same way in two different cultural contexts (and in two different languages). At the very least, this issue should be commented upon. Better still would be to use some method of evaluating equivalence – there are some straightforward methods available for smaller sample sizes (e.g., Mantel-Haenszel statistics, Ordinal Logistic Regression). While the authors do address equivalence issues in the discussion to a certain extent, these appear to refer more to patterns of findings across the two groups.

2. It appears that several of the analyses involve predicting a subscore of the BDI-II while controlling for the total score. This approach is problematic, as it means that the same items are contributing fully to the DV score and partially to the covariate score. Covariates, IVs, and DVs should really be statistically independent from one another. One solution would be to control in each case for the other subscale, rather than controlling the total score. This would remove variance shared between the two subscales (which could be interpreted as representing ‘general depression’).

Additional comments

1. In the introduction, the authors make a distinction between earlier studies that emphasized non-Western somatization with more recent studies that emphasize Western psychologization. While I believe this is largely accurate, I’m not sure it’s fair to say that these are inconsistent findings – rather, what we have is a change in emphasis. Earlier findings could have been interpreted as supporting Western psychologization, but were constrained by the assumption that cultural phenomena worth reporting are those that differ from the Western ‘norm’. Similarly, more recent studies continue to find evidence for somatization, but suggest that the evidence for Western psychologization is stronger and more compelling. Note that Parker’s study, cited later, could also be included in the list of studies that have tended to find better evidence for psychologization.

2. On lines 84-85, the authors note that, “somatization is the norm within Chinese populations (Ryder, Yang, & Heine, 2002)”. But earlier on the same page, they argue that one should not over-interpret this finding, as it does not apply to anxiety. If so, the later statement should be appropriately qualified. I also wonder how the stigma-avoidance explanation for somatization fits with the finding of more somatic symptom reporting in Western patients with anxiety.

3. The authors cite Dere et al (2013) to support the idea that ‘Chinese somatization’ is not a straightforward finding – European Americans appear to have higher levels of reversed somatic symptoms (e.g., hypersomnia, hyperphagia). As such, these findings demand qualification of sentences like that on lines 97-98: “Additionally, a somatic emphasis in the Chinese culture…”

4. In the methods, the authors report overall reliability for the BDI-II in the two samples. Since reliability is a property of scores rather than scales, reliabilities should be presented for every score used in the paper. In other words, we should see estimates for the two BDI-II subscales as well as the total score.

5. In the discussion, the authors note that the sample sizes are small but also argue that they are sufficient to detect main and interaction effects. On what grounds is this argument made? One cannot have sufficient power to detect ‘effects’ – one can only have sufficient power to detect a given effect size. What kinds of effect sizes were the authors looking for, how did they define ‘sufficient power’, and to what extent did they have such power to find effects of that size?

---

## Round 0.2 · accepted · Accept

I have reviewed your revision and I am happy to Accept the manuscript.